# Analysis of Heart Rate Variability in Individuals Affected by Amyotrophic Lateral Sclerosis

**DOI:** 10.3390/s24072355

**Published:** 2024-04-07

**Authors:** Rosa Maset-Roig, Jordi Caplliure-Llopis, Nieves de Bernardo, Jesús Privado, Jorge Alarcón-Jiménez, Julio Martín-Ruiz, Marta Botella-Navas, Carlos Villarón-Casales, David Sancho-Cantus, José Enrique de la Rubia Ortí

**Affiliations:** 1Doctoral Degree School, Catholic University San Vicente Mártir, 46001 Valencia, Spain; rosamaria.maset@ucv.es; 2Department of Nursing, Catholic University San Vicente Mártir, 46001 Valencia, Spain; jordi.caplliure@ucv.es (J.C.-L.); marta.botella@ucv.es (M.B.-N.); joseenrique.delarubi@ucv.es (J.E.d.l.R.O.); 3Department of Physiotherapy, Catholic University San Vicente Mártir, 46001 Valencia, Spain; nieves.debernardo@ucv.es (N.d.B.); jorge.alarcon@ucv.es (J.A.-J.); 4Department of Methodology of Behavioral Sciences, Universidad Complutense de Madrid, Campus de Somosaguas, Pozuelo de Alarcón, 28223 Madrid, Spain; jesus.privado@pdi.ucm.es; 5Department of Health and Functional Evaluation, Faculty of Physical Activity and Sports Sciences, Catholic University San Vicente Mártir, 46001 Valencia, Spain; julio.martin@ucv.es; 6Biomechanics & Physiotherapy in Sports (BIOCAPS), Faculty of Health Sciences, Spain, European University of Valencia, 46001 Valencia, Spain

**Keywords:** heart rate variability, amyotrophic lateral sclerosis, autonomic nervous system

## Abstract

Introduction: Amyotrophic lateral sclerosis (ALS) produces alterations in the autonomic nervous system (ANS), which explains the cardiac manifestations observed in patients. The assessment of heart rate variability (HRV) is what best reflects the activity of the ANS on heart rate. The Polar H7 Bluetooth^®^ device proves to be a non-invasive and much faster technology than existing alternatives for this purpose. Objective: The goal of this study is to determine HRV using Polar H7 Bluetooth technology in ALS patients, comparing the obtained measurements with values from healthy individuals. Method: The sample consisted of 124 participants: 68 diagnosed with ALS and 56 healthy individuals. Using Polar H7 Bluetooth technology and the ELITE HRV application, various HRV measurements were determined for all participants, specifically the HRV index, RMSSD, RMSSD LN, SDNN index, PNN50, LF, HF, LF/HF ratio, HR average, and HF peak frequency. Results: Statistically significant differences were observed between ALS patients and healthy individuals in the HRV index, RMSSD, RMSSD LN, SDNN index, PNN50, HF, and LF, where healthy individuals exhibited higher scores. For the HR average, the ALS group showed a higher value. Values were similar when comparing men and women with ALS, with only a higher HF peak frequency observed in women. Conclusion: The Polar H7 Bluetooth^®^ device is effective in determining heart rate variability alterations in ALS, being a promising prognostic tool for the disease.

## 1. Introduction

Amyotrophic lateral sclerosis (ALS) is a neurodegenerative disease of the central nervous system that affects motor neurons, leading to the paralysis of voluntary muscles [1]. It is a rare disease with an annual incidence of 1.4 cases per 100,000 inhabitants and a higher prevalence in males [2]. ALS is incurable, and life expectancy ranges from 2 to 5 years [3,4]. Bulbar ALS is the most severe due to its progression and level of impairment. However, in both bulbar and spinal types, the disease’s progression ultimately results in cardiac and respiratory issues as the primary causes of mortality [4,5]. The proper functioning of these two systems depends on the activity of the autonomic nervous system (ANS). The ANS comprises the sympathetic nervous system (SNS) and the parasympathetic nervous system (PNS). The SNS and PNS work antagonistically, with certain differences between men and women [6,7], regulating, among other functions, heart rate and rhythm [5]. In ALS patients, alterations in the ANS have been observed even in the early stages of the pathology [8,9]. Signs of autonomic dysfunction are based on an imbalance between sympathetic (S) and parasympathetic (PS) innervation, as there is a decrease in PS control and increased S activity [10]. In this context, HRV [11] is the most reflective measure of the combined activity of the SNS and PNS on heart rate. Therefore, HRV is an excellent indicator of ANS integrity and an effective prognostic factor [12], with lower HRV predicting a higher risk of heart disease, sudden death, and various types of mortality [13]. Nevertheless, alterations in both HRV and other variables indicating abnormal ANS activity in ALS patients have not been thoroughly investigated in comparison to healthy individuals. Furthermore, assessments conducted to evaluate nervous function to date have either been invasive or required sustained control over time [12]. In this context, the Polar H7 Bluetooth^®^ device connected to a chest belt, following the protocols of Moya (2022), enables data analysis through the ELITE HRV application in a 60 s timeframe. Hence, this technology is non-invasive and considerably faster than existing methods, with measurement times lasting up to 24 h [14], such as the use of Holter monitoring [10,15,16] or high-resolution ultrasound studies [17].

Therefore, the aim of this study was to determine the effectiveness and suitability of Polar H7 Bluetooth technology to measure the HRV of ALS patients, taking a sample of healthy subjects as a reference.

## 2. Materials and Methods

### 2.1. Participants

To obtain the population sample of patients with ALS, contact was made with the main Spanish ALS associations, sending the project to their directors, who in turn passed it on to their affiliates. Volunteers who wished to participate contacted the main project researcher, who informed them through a personal interview and an information sheet and signed the informed consent. To obtain healthy volunteers, mainly workers from these associations or collaborators in the study were included, and they were also informed about the nature of the project and provided their consent. From these volunteers (patients with ALS and healthy individuals), the selection was made by applying the following selection criteria (Figure 1): males over 18 years, non-fertile females over 50, or females between 18 and 50 years old; patients diagnosed with and symptomatic of ALS for at least 6 months prior to study inclusion; patients undergoing Riluzole treatment; and willingness to participate in the study by signing the informed consent form. Exclusion criteria were as follows: patients with tracheotomy, patients with invasive or non-invasive positive pressure ventilation, participation in any other trial in the 4 weeks before inclusion; individuals with dementia, individuals with alcohol or drug abuse, patients infected with hepatitis B or C or who are HIV positive, renal patients with creatinine levels twice as high as normal markers 30 days before inclusion, and hepatic patients with elevated hepatic markers (ALT, AST) three times above normal 30 days before inclusion. In the case of healthy subjects, inclusion criteria were as follows: individuals of both sexes over 18 years old without a history of cardiac diseases or autonomic nervous system disorders and willingness to participate in the study by signing the informed consent form. Exclusion criteria were as follows: participation in any other trial in the 4 weeks before inclusion, individuals with dementia, and individuals with alcohol or drug abuse.

Finally, the sample consisted of 124 participants: 68 individuals diagnosed with amyotrophic lateral sclerosis (ALS) and 56 healthy subjects. The ALS patients had a mean age of 56.74 years (SD = 10.47); 61.8% were male, and 80.9% had spinal ALS, in line with the demographic characteristics of the disease [4]. Healthy participants had a mean age of 56.89 years (SD = 10.37 years), with an equal distribution of genders.

### 2.2. Materials and Instruments

A Polar H7 Bluetooth^®^ device was used to collect the data. This instrument measures heart rate and can be connected to Bluetooth-enabled devices compatible with heart rate measurement software. A specific application is required to view the heart rate data on the receiving device. Additionally, an independent application called Elite HRV was utilized to measure heart rate variability.

### 2.3. Design

An analytical cross-sectional study was conducted, comparing HRV between patients affected by ALS and healthy individuals. The following measurements were considered for the analysis of HRV:

Temporal variables:

HRV index: It represents the variation over time between RR intervals in the electrocardiogram, defined as the physiological variation in the duration of the interval between each heartbeat [13,18,19,20].

RMSSD: The root mean square of successive differences is calculated as the square root of the mean sum of the squared differences between adjacent RR intervals. In HRV, it is primarily associated with the activity of the nervous system, indicating that a higher index reflects better regulation of the ANS on heart rhythm. It particularly reflects the PNS or vagal influence on the heart [13,19].

RMSSD LN: A natural logarithm is applied to RMSSD to distribute the figures into a more easily understandable range. Similar to RMSSD, it primarily reflects PNS activity [13,20].

SDNN: It is the standard deviation of all RR intervals. SDNN reflects the combined modulation of the sympathetic and parasympathetic branches of the ANS on the sinus node. Higher values indicate a greater capacity of the ANS to modulate heart rhythm under different situations, associated with a better prognosis [13].

PNN50: It is the percentage of consecutive RR intervals differing by more than 50 ms. PNN50 primarily measures the influence of the PNS through HRV analysis by quantifying very rapid oscillations of the rhythm mediated by the vagal system [19,21].

Frequency variables:

Heart Rate (HR): This refers to the number of heartbeats per minute. Variations in frequency, whether high or low, correlate with the sympathetic–parasympathetic balance. HF reflects the PSNS; LF reflects the SNS and/or PSNS [13,18].

The following are the variables that can be obtained:

LF: Low-frequency heart rate reflects the combined activity of the SNS and PSNS. Sympathetic mechanisms predominate, but there is also some parasympathetic modulation [19,20,22,23].

HF: High-frequency heart rate primarily represents the activity of the PSNS or vagus nerve on the heart. The higher the HF index, the greater the cardiac vagal tone [19,20,22,23].

LF/HF Ratio: This is the ratio between LF and HF heart rate. This ratio allows for the non-invasive evaluation of the autonomic cardiac control balance. A higher ratio indicates sympathetic predominance [23].

HF Peak Frequency: This refers to the high-frequency component, the maximum heart rate reached during physical exercise, which falls within the range of 0.15–0.4. It mainly represents the activity of the SNS [21].

HR Average: It is the average of the heart rate, referring to the arithmetic mean of all RR intervals (time in milliseconds between two consecutive heartbeats) obtained in an HRV recording. Therefore, HR average allows for a simple estimation of the average speed at which the heart beats during an HRV analysis. This variable can indicate alterations in the sympathetic–parasympathetic system [13,20,24].

### 2.4. Procedure

The tests were conducted in a classroom with an average temperature of 25 °C and a humidity level ranging from 50% to 60%. To ensure data reproducibility, the tests were performed prior to the initial data collection. During the initial five-minute adaptation period, the H7 sensor was positioned below the xiphoid process of each participant. Data collection took place for one minute using the Moya Ramon protocol with patients in a seated position. Before placing the monitor, participants were informed about the importance of not speaking or moving during the measurements. It was also indicated that respirations should be spontaneous during the measurement, avoiding controlled breathing as it can artificially increase parasympathetic measurements of HRV [25]. Data analysis was carried out utilizing the ELITE HRV application connected to the Polar H7 chest strap (Figure 2).

### 2.5. HRV Analyses

The Elite HRV application receives heart rate data from the Polar H7 Bluetooth sensor via Bluetooth Smart, with a heart rate sampling frequency that can be set to 1 Hz, 2 Hz, 4 Hz, or 5 Hz. Data processing involves preliminary cleaning to remove artifacts and noises that may interfere with the results, allowing for up to 5% as they can greatly influence the outcomes [26]. Subsequently, intervals of heart rate that are either too short or too long are removed, and various HRV metrics are calculated. HRV data can be visualized in the form of graphs and trends, and personalized reports are generated providing an interpretation of the HRV data.

### 2.6. Data Analysis

First, descriptive statistics for the main variables of HRV and age were computed, along with their distribution. Second, differences between ALS patients and healthy individuals in terms of age and HRV measurements were investigated using an independent samples *t*-test for variables with a normal distribution and a non-parametric Mann–Whitney U test for those not conforming to normal distribution. A chi-square test was employed to assess differences in gender proportions between ALS patients and healthy individuals. Third, a *t*-test for normally distributed measurements and a Mann–Whitney U test for non-normally distributed measurements were used to determine if there were gender-related differences in HRV means. Finally, normative values for HRV measurements in ALS patients were calculated. All analyses were conducted using the statistical package SPSS V. 25.

### 2.7. Ethical Concerns

The project was approved by the Clinical Research Ethics Committee of Hospital de la Fe in Valencia, Spain (2021-001989 38), and was conducted in accordance with the principles outlined in the Declaration of Helsinki.

## 3. Results

Table 1 displays the descriptive statistics and distribution (skewness and kurtosis) for the age of participants and various HRV measures. It is evident that the majority of the measurements do not exhibit a normal distribution, showing in most cases a positively skewed and leptokurtic distribution. Skewness values exceeding 2 (in absolute value) and kurtosis values surpassing 7 (in absolute value) indicate a significant deviation from normality [27].

### 3.1. Differences in Cardiac Variability between ALS Patients and Healthy Individuals

A first assessment was conducted to determine if the two participant groups (ALS patients and healthy individuals) exhibited differences in terms of age and gender, ensuring equivalence in these variables. As age followed a normal distribution, an independent samples *t*-test was performed, revealing no statistically significant differences between the means for ALS and healthy individuals (t122 = 0.08, *p* = 0.993). As gender was a qualitative measure, a chi-square test was employed, indicating no differences in the proportion of each gender in both groups (χ^2^(1) = 1.73, *p* = 0.207), with a very low coefficient of contingency (C = 0.117, *p* = 0.189). Therefore, the two groups were matched in terms of age and gender, allowing for the comparison of HRV between the ALS and healthy groups without considering age and gender as relevant variables.

Second, to verify if there were differences between the two participant groups in HRV, an independent samples *t*-test was utilized if the variable exhibited a normal distribution. Alternatively, a non-parametric independent samples test, the Mann–Whitney U test, was employed if the variable did not follow a normal distribution. Table 2 presents the obtained results.

Statistically significant differences were observed across all HRV measurements, except for the LF/HF ratio and HF peak frequency. In all measurements, except the HR average, healthy individuals exhibited higher scores than the ALS group. Additionally, the effect size of the standardized mean difference (Cohen’s d) was calculated, revealing values between medium and high for most differences. According to Cohen, a d ≤ 0.20 is considered small, d = 0.50 is medium, and d ≥ 0.80 is considered large [28].

### 3.2. Sex Differences in Cardiac Variability in ALS

Table 3 displays the differences in HRV between both sexes for ALS patients. Statistically significant differences were found only for the HF peak frequency, with higher scores for females, exhibiting a medium-to-high effect size. For the remaining measurements, there were no statistically significant differences, and the effect size was medium to low.

### 3.3. Reference Values for Heart Rate Variability in ALS patients

Table 4 presents the reference values for heart rate variability in ALS patients. Specifically, for each raw score, the percentile score, standardized score (Z), normalized Z, and T score (*M* = 50, *SD* = 10) were calculated.

## 4. Discussion

The main objective of our study was to analyze the HRV in individuals affected by ALS and compare the results with a healthy control group of the same age range.

For this analysis, variables were grouped into temporal measurements on one hand and into frequency measurements on the other.

Concerning the analysis of temporal measurements, our findings confirmed significantly lower HRV index values in the ALS group compared to healthy patients. These results are aligned with those obtained by various authors, such as Pinto et al. [29], who studied a population of ALS patients with a mean age of 57 years for 4 h; Silveira et al. [30], who measured HRV for 20 min; and Weisy et al. [17], who measured it for 10 min. These diminished HRV index values suggest poor adaptation of the ANS, representing a significant health risk factor.

Additionally, statistically significant lower values of the RMSSD were found in the ALS group compared to the healthy group. Similar results were also observed by authors such as Pavlovic et al. [10] and Iscan et al. [15], both of whom conducted Holter monitoring for 24 h on 55 patients with a mean age of 56 and 59.8 years, respectively. These outcomes may suggest reduced vagal activity associated with cardiac electrical activity [30]. An RMSSD below 10 ms, coupled with a decrease in SDNN below 20 ms, is associated with a high risk of developing cardiac diseases.

Similarly, the values obtained for the RMSSD LN are also statistically significantly lower in patients with ALS, consistent with observations made by Moore [31]. This outcome could suggest a high risk of developing cardiac diseases and significant dysfunction of the ANS in ALS [13,32].

In terms of SDNN, values within the normal range in our healthy group were observed, but these were significantly lower in the ALS group (30.07 ms). This aligns with the findings of Brown et al. [33], who, after measuring for 3 days, reported similar results. Values below 35 are considered a high health risk due to ANS weakness [16], and an SDNN interval below 19 ms is associated with imminent death within a year [24].

The PNN50 is directly associated with the activity of the ANS; values around 3% are considered to be at the lower limit, representing a high health risk. Our results in ALS patients are slightly above this threshold but significantly lower than those in the healthy group, indicating a high health risk in these patients. This finding is consistent with descriptions by other authors [10,15,17,30,34].

Regarding the analysis of frequency measurements, the variable LF exhibits statistically significantly lower values in the ALS group compared to the healthy group. An LF is associated with sympathetic activity, in a way that low LF values indicate an increase in parasympathetic activity and, consequently, a predominance of slow and gentle breathing, reflecting a state of patient relaxation [24]. Consequently, these extremely low values could indicate a parasympathetic dominance associated with states of fatigue [24]. It is worth noting that our results align with the findings of other authors [15,30,35] but differ slightly from other studies that observed somewhat lower values [36,37].

The variable HF is sometimes referred to as “respiratory bands” and is associated with respiratory rhythm, reflecting parasympathetic activity. In our study, we observed that the ALS group exhibits statistically significantly lower values than the healthy group, in line with other studies [15,30,35,36,37], suggesting a decrease in parasympathetic activity, possibly associated with the presence of stress, panic, or anxiety [29]. Furthermore, these lower values could be explained by the loss of neural control over the cardiovascular system, particularly a reduction in vagal tone [20].

The LF/HF ratio variable is related to the balance of the ANS, where values above 2 would indicate a predominance of the SNS [38]. Our results, both in the ALS and healthy groups, surpass these values, aligning with findings from other authors [34,35,36]. This suggests a predominance of the SNS in both populations.

The HF peak frequency variable is associated with the ANS, where a higher HF peak is linked to a state of relaxation, while a lower value indicates a state of stress or alertness, consequently activating the SNS [21]. Our results did not show significant differences between ALS patients and healthy subjects. This variable, to the best of our knowledge, has not been assessed in previous studies, making it a novel contribution. In the context of ALS, it reinforces the presence of a decrease in vagal tone and a predominance of the SNS, as the values fall within the normal range but are closer to the lower limit.

The variable HR average, which is associated with alterations in the regulation of the ANS, has also been analyzed, correlating high/low values of the HR average with sympathetic/vagal tone [24]. Our results reveal a significantly higher value in patients with ALS, potentially indicating a predominance of sympathetic activity within the ALS population, in line with the disturbances observed in other HRV variables already discussed.

However, it is worth noting that the increase in the sympathetic activity of the ANS is a main effector of the stress response system and is responsible for the cardiovascular reaction to episodes of emotional, social, and physical stress, characterized by a pressor response and an increase in heart rate [39]. Therefore, our findings may indicate that there is a mechanism of resilience to stress in patients with ALS, aiming to promote their survival.

All these findings, obtained through the use of the Polar H7 Bluetooth^®^ device and which indicate sympathetic dysfunction in ALS, may elucidate the myocardial structural defects observed in cardiac magnetic resonance imaging and, consequently, the reported cardiac deaths in late-stage ALS patients [38]. In this context, it is crucial to establish reference values obtained with the Polar H7 Bluetooth^®^ device. Therefore, in the results section (Table 4), we provide benchmarks for HRV. These data will enable future evaluators to have reference values to aid in determining the cardiac status of ALS patients, potentially serving as a diagnostic tool for assessing disease progression.

To complete the HRV analysis, building upon the sex-based differences observed in the ANS in healthy individuals, efforts were made to ascertain potential distinctions between men and women with ALS. This analysis revealed variations solely in the frequency HF peak, with higher scores observed in women, demonstrating a medium-to-high effect size. No other variables exhibited sex-based differences in patients, maintaining the same distribution observed in healthy individuals within our study and consistent with findings by authors such as Moore et al. [31], who did not find sex-based differences either. Nevertheless, the impact of gender on HRV in ALS patients remains largely unexplored in the literature and should be taken into consideration for future research.

## 5. Conclusions

In conclusion, our findings indicate that the Polar H7 Bluetooth^®^ device proves effective in determining HRV, revealing reduced sympathetic and parasympathetic activity in ALS patients compared to healthy individuals, albeit with a greater sympathetic predominance in the ALS population. Notably, this device is a less costly, less cumbersome, and quicker tool compared to other technologies employed to date. Thus, the Polar H7 Bluetooth^®^ device emerges as a promising prognostic tool for the disease.

However, it is important to acknowledge certain limitations in this study, such as the sample variability, given the rarity of the disease and, in some cases, its rapid progression. This constraint impeded obtaining a larger and more homogenous population sample and grouping it by time elapsed since diagnosis. Furthermore, the fact that it is such a heterogeneous population could partly explain the high dispersion of the results, indicating that a personalized approach might be more adequate than conducting measurements over a minute.

## Figures and Tables

**Figure 1 sensors-24-02355-f001:**
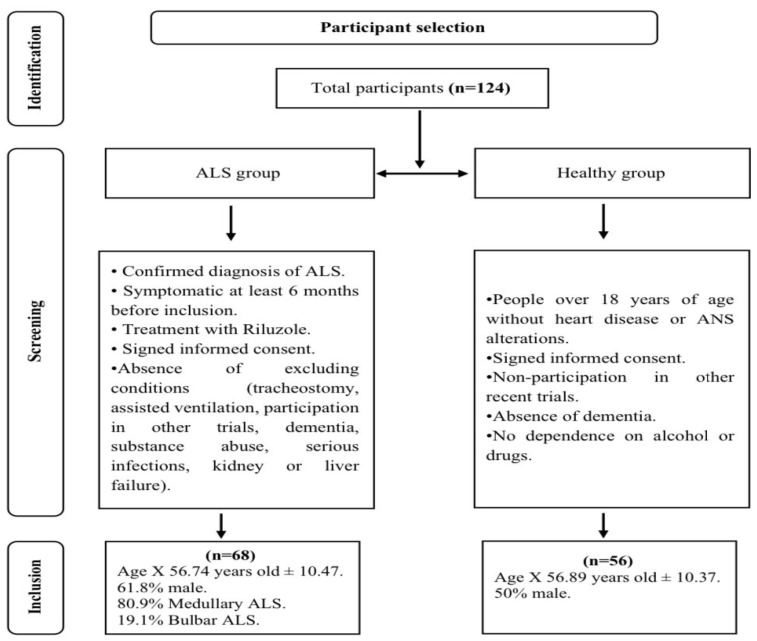
Selection of the study population sample.

**Figure 2 sensors-24-02355-f002:**
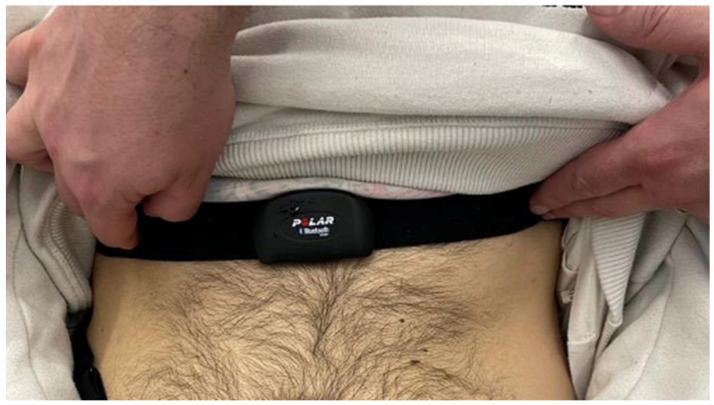
Polar H7 Bluetooth^®^ device.

**Table 1 sensors-24-02355-t001:** Descriptive statistics, Skewness, and Kurtosis of Age and Heart Rate Variability.

	M	SD	Skewness	Kurtosis
Age	56.81	10.39	−0.13	−0.39
HRV index	46.70	12.77	0.16	0.20
HRV RMSSD ms	30.87	30.43	2.60 *	8.09 *
HRV LN RMSSD ms	3.08	0.84	0.04	0.10
HRV SDNN ms	42.81	38.80	2.96 *	10.91 *
HRV PNN50	8.56	13.97	2.39 *	6.17 *
LF Power ms2	833.47	1703.79	3.50 *	13.07 *
HF Power ms2	633.23	1847.79	4.63 *	22.99 *
LF HF ratio	4.07	5.59	3.26 *	13.18 *
HF Peak HZ	0.26	0.09	0.77	−0.07
HR Average	76.66	13.73	0.80	0.3

* Measurements with a Non-Normal Distribution.

**Table 2 sensors-24-02355-t002:** Differences in Means for Cardiac Variability between ALS Patients and Healthy individuals.

	Group	M	SD	Differences in Means	*d* Cohen
HRV index	ALS	42.33	12.21	*t*_123_ = 4.7, *p* < 0.001	0.82
	Healthy	52.07	11.39		
HRV RMSSD ms	ALS	22.00	22.28	*U* = 952.00, *p* < 0.001	0.86
	Healthy	41.65	35.37		
HRV LN RMSSD ms	ALS	2.76	0.81	*t*_121_ = 5.14, *p* < 0.001	0.93
	Healthy	3.47	0.71		
HRV SDNN ms	ALS	30.07	25.28	*U* = 81.50, *p* < 0.001	0.99
	Healthy	58.28	46.30		
HRV PNN50	ALS	4.79	10.57	*U* = 1014.00, *p* < 0.001	0.84
	Healthy	13.14	16.17		
Frequency LF Power ms2	ALS	461.62	1231.02	*U* = 1014.00, *p* < 0.001	0.81
	Healthy	1285.00	2065.76		
Frequency HF Power ms2	ALS	366.00	1250.29	*U* = 1067.00, *p* < 0.001	0.76
	Healthy	957.72	2352.85		
Frequency LF HF ratio	ALS	3.94	5.48	*U* = 1792.00, *p* = 0.574	0.10
	Healthy	4.22	5.77		
Frequency HF Peak HZ	ALS	0.27	0.09	*t*_122_ = 0.98, *p* = 0.330	0.18
	Healthy	0.25	0.08		
HR Average	ALS	79.30	14.11	*t*_123_ = 2.24, *p* = 0.016	0.44
	Healthy	73.41	12.62		

**Table 3 sensors-24-02355-t003:** Sex differences in Heart Rate Variability in ALS patients.

	Group	M	SD	Differences in Means	*d* Cohen
HRV index	Male	41.12	11.64	*t*_66_ = 0.78, *p* = 0.440	0.20
	Female	43.48	12.49		
HRV RMSSD ms	Female	18.70	12.37	*U* = 512.00, *p* = 0.866	0.04
	Male	24.35	26.50		
HRV LN RMSSD ms	Female	2.69	0.77	*t*_64_ = 0.67, *p* = 0.508	0.17
	Male	2.82	0.83		
HRV SDNN ms	Female	24.90	15.85	*U* = 425.00, *p* = 0.195	0.33
	Male	33.67	29.25		
HRV PNN50	Female	3.32	7.05	*U* = 487.00, *p* = 0.575	0.14
	Male	5.79	12.28		
Frequency LF Power ms2	Female	195.69	284.21	*U* = 396.00, *p* = 0.094	0.42
	Male	629.76	1533.91		
Frequency HF Power ms2	Female	134.35	230.96	*U* = 446.00, *p* = 0.306	0.26
	Male	512.41	1570.20		
Frequency LF HF ratio	Female	3.16	3.09	*U* = 466.50, *p* = 0.448	0.19
	Male	4.35	6.55		
Frequency HF Peak HZ	Female	0.31	0.09	*t*_65_ = 3.06, *p* = 0.003	0.77
	Male	0.24	0.08		
HR Average	Female	81.92	14.43	*t*_66_ = 1.34, *p* = 0.186	0.34
	Varón	77.26	13.67		

**Table 4 sensors-24-02355-t004:** Reference Values for Heart Rate Variability in ALS patients.

		Centile			Z_n_		T			HRV Index		Z_HRV_ Index		HRV RMSSD ms		Z_RMSSD ms_		HRV LN RMSSD ms		Z_HRV LN RMSSD ms_		HRV SDNN ms			Z_SDNN ms_		HRV PNN50		Z_HRV PNN50_	
		1			−2.33		26.70			9.00		−2.73		1.82		−0.91		0.60		−2.67		5.74			−0.96		0.00		−0.45	
		5			−1.64		33.60			23.00		−1.58		4.45		−0.79		1.45		−1.61		8.23			−0.86		0.00		−0.45	
		10			−1.28		37.20			26.00		−1.34		5.52		−0.74		1.71		−1.30		10.01			−0.79		0.00		−0.45	
		15			−1.04		39.60			30.50		−0.97		7.10		−0.67		1.94		−1.01		11.86			−0.72		0.00		−0.45	
		20			−0.84		41.60			32.00		−0.85		8.09		−0.62		2.09		−0.83		13.64			−0.65		0.00		−0.45	
		25			−0.67		43.30			35.00		−0.60		9.84		−0.55		2.24		−0.64		14.98			−0.60		0.00		−0.45	
		30			−0.52		44.80			36.00		−0.52		10.49		−0.52		2.32		−0.54		16.30			−0.54		0.00		−0.45	
		35			−0.39		46.10			38.00		−0.35		12.25		−0.44		2.47		−0.35		18.31			−0.47		0.00		−0.45	
		40			−0.25		47.50			39.00		−0.27		12.70		−0.42		2.54		−0.27		20.32			−0.39		0.00		−0.45	
		45			−0.13		48.70			39.50		−0.23		13.21		−0.39		2.57		−0.23		22.54			−0.30		0.00		−0.45	
		50			0.00		50.00			41.00		−0.11		14.78		−0.32		2.68		−0.10		23.85			−0.25		0.00		−0.45	
		55			0.13		51.30			43.00		0.05		16.12		−0.26		2.80		0.04		25.81			−0.17		0.00		−0.45	
		60			0.25		52.50			44.00		0.14		17.61		−0.20		2.87		0.14		28.30			−0.07		0.00		−0.45	
		65			0.39		53.90			45.50		0.26		19.98		−0.09		3.03		0.34		31.00			0.04		1.00		−0.36	
		70			0.52		55.20			49.00		0.55		24.79		0.13		3.23		0.58		33.59			0.14		3.30		−0.14	
		75			0.67		56.70			51.00		0.71		27.98		0.27		3.33		0.70		34.69			0.18		4.75		0.00	
		80			0.84		58.40			52.00		0.79		29.29		0.33		3.38		0.77		38.50			0.33		7.20		0.23	
		85			1.04		60.40			54.50		1.00		34.76		0.57		3.55		0.97		44.04			0.55		10.00		0.49	
		90			1.28		62.80			58.00		1.28		43.98		0.99		3.78		1.26		57.49			1.08		15.40		1.00	
		95			1.64		66.40			66.50		1.98		75.34		2.39		4.33		1.93		70.64			1.60		32.10		2.58	
		99			2.33		73.30			66.50		1.98		75.34		2.39		4.33		1.93		70.64			1.60		32.10		2.58	
Mean					0.00		50.00			42.33		0.00		22.00		0.00		2.76		0.00		30.07			0.00		4.79		0.00	
DT					1.00		10.00			12.21		1.00		22.28		1.00		0.81		1.00		25.28			1.00		10.57		1.00	
	**Centile**		**Z_n_**	**T**		**Frequency LF Power ms2**		**Z _Frequency LF Power ms2_**	**Frequency HF Power ms2**		**Z _Frequency HF Power ms2_**		**Frequency LF HF ratio**		**Z _Frequency LF HR ratio_**		**Frequency HF Peak HZ**		**Z _Frequency HF Peak Hz_**		**Frequency HF Peak HZ (Female)**		**Z _Frequency HF Peak Hz (Female)_**	**Frequency HF Peak HZ (Male)**		**Z _Frequency HF Peak Hz (Male)_**		**HR Average**		**Z_HR Average_**
	1		−2.33	26.70		7.70		−0.37	0.91		−0.29		0.08		−0.70		0.16		−1.24		0.16		−1.08	0.17		−1.52		54.00		−1.79
	5		−1.64	33.60		14.18		−0.36	3.82		−0.29		0.25		−0.67		0.16		−1.24		0.16		−1.08	0.17		−1.50		58.00		−1.51
	10		−1.28	37.20		21.18		−0.36	7.94		−0.29		0.57		−0.62		0.16		−1.24		0.16		−1.08	0.18		−1.38		64.00		−1.08
	15		−1.04	39.60		26.44		−0.35	12.21		−0.28		0.73		−0.59		0.17		−1.06		0.16		−1.08	0.20		−1.19		66.00		−0.94
	20		−0.84	41.60		35.64		−0.35	16.09		−0.28		0.98		−0.54		0.19		−0.90		0.16		−1.08	0.25		−0.66		68.00		−0.80
	25		−0,67	43.30		43.88		−0.34	20.88		−0.28		1.08		−0.52		0.19		−0.86		0.17		−0.88	0.25		−0.66		69.00		−0.73
	30		−0.52	44.80		64.31		−0.32	25.57		−0.27		1.24		−0.49		0.20		−0.71		0.19		−0,68	0.25		−0.66		70.00		−0.66
	35		−0.39	46.10		74.52		−0.31	27.94		−0.27		1.59		−0.43		0.22		−0.54		0.20		−0.598	0.27		−0.48		72.50		−0.48
	40		−0.25	47.50		82.61		−0.31	34.80		−0.26		1.82		−0.39		0.22		−0.54		0.21		−0.45	0.27		−0.48		74.00		−0.38
	45		−0.13	48.70		98.02		−0.30	38.97		−0.26		2.07		−0.34		0.25		−0.19		0.22		−0.29	0.28		−0.36		75.50		−0.27
	50		0.00	50.00		124.13		−0.27	43.20		−0.26		2.40		−0.28		0.25		−0.19		0.22		−0.29	0.30		−0.13		76.00		−0.23
	55		0.13	51.30		145.97		−0.26	59.90		−0.24		2.65		−0.24		0.27		−0.02		0.22		−0.29	0.30		−0.08		78.50		−0.06
	60		0.25	52.50		172.95		−0.23	77.65		−0.23		2.88		−0.19		0.27		−0.01		0.25		0.10	0.31		0.04		81.00		0.12
	65		0.39	53.90		205.63		−0.21	10.79		−0.20		3.43		−0.09		0.28		0.15		0.25		0.10	0.33		0.19		83.00		0.26
	70		0.52	55.20		253.59		−0.17	124.83		−0.19		3.80		−0.03		0.30		0.38		0.27		0.30	0.35		0.46		86.00		0.47
	75		0.67	56.70		308.25		−0.12	155.01		−0.17		4.26		0.06		0.33		0.67		0.29		0.53	0.38		0.82		88.50		0.65
	80		0.84	58.40		400.82		−0.05	200.64		−0.13		5.98		0.37		0.34		0.85		0.33		1.15	0.39		0.91		91.00		0.83
	85		1.04	60.40		595.04		0.11	279.69		−0.07		8.38		0.81		0.37		1.13		0.34		1.27	0.43		1.29		92.00		0.90
	90		1.28	62.80		995.53		0.43	467.56		0.08		8.96		0.92		0.39		1.41		0.36		1.46	0.47		1.74		98.00		1.33
	95		1.64	66.40		2486.67		1.65	1947.23		1.26		9.75		1.06		0.47		2.25		0.42		2.17	0.48		1.94		109.00		2.10
	99		2.33	73.30		2486.67		1.65	1947.23		1.26		9.75		1.06		0.47		2.25		0.42		2.17	0.48		1.94		109.00		2.10
Mean			0.00	50.00		461.62		0.00	366.00		0.00		3.94		0.00		0.27		0.00		0.24		0.00	0.31		0.00		79.30		0.00
DT			1.00	10.00		1231.02		1.00	1250.29		1.00		5.48		1.00		0.09		1.00		0.08		1.00	0.09		1.00		14.11		1.00

## Data Availability

Data are contained within the article.

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
