# Peer review of "Analysis of Heart Rate Variability in Individuals Affected by Amyotrophic Lateral Sclerosis"

_sensors, 2024, doi:10.3390/s24072355_

Round 1
Reviewer 1 Report
Comments and Suggestions for Authors
This study aims to investigate alteration in the autonomic nervous system (ANS) in patients with Amyotrophic Lateral Sclerosis (ALS) using heart rate variability (HRV) analysis. The authors performed HRV analysis on 68 ALS patients and compared the results with 56 controls. The authors reported statistically significant alteration in HRV indexes from ALS patients, but this article has several concerns to be addressed.
1. Description about their experiment, especially data acquisition and processing, are too meager to provide reproducibility. The “data analysis” section doesn’t contain any description about HR processing and HRV analysis. It only contains description about statistics.
2. Likewise, the authors say that “Data collection took place for one minute using the Moya Ramon protocol.” Is the protocol proper to compare HRV alteration in ALS patients?
3.Is one minute interval enough to perform frequency-domain HRV analysis without any special processing technique? Generally, in order to measure power for LF band (0.04 to 0.15) within a short interval less than five minutes, a highly skilled analysis, as known as (ultra) short term HRV analysis, is required. This maybe the reason the author failed to find significant difference in frequency domain analysis using LH/HF ration, which can clearly shows the balance of the Sympathetic Nervous System (SNS) and the Parasympathetic Nervous System (PNS).
Comments on the Quality of English Language
Quality of English is fair, but the descriptions are insufficient to comprehend details and support results.
Author Response
The authors presented the work titled as "Analysis of Heart Rate Variability in Individuals Affected by Amyotrophic Lateral Sclerosis."
There are some comments which need to be addressed:
This study aims to investigate alteration in the autonomic nervous system (ANS) in patients with Amyotrophic Lateral Sclerosis (ALS) using heart rate variability (HRV) analysis. The authors performed HRV analysis on 68 ALS patients and compared the results with 56 controls. The authors reported statistically significant alteration in HRV indexes from ALS patients, but this article has several concerns to be addressed.
- Description about their experiment, especially data acquisition and processing, are too meager to provide reproducibility. The “data analysis” section doesn’t contain any description about HR processing and HRV analysis. It only contains description about statistics.
We fully agree with the reviewer that this information was missing, which was necessary for the reproducibility of the assay and that has been added in the Materials and Methods section (completing point 2.4. and adding point 2.5.). Section 2.1. has also been completed, related to the procedure for obtaining the study population sample of ALS patients and healthy individuals.
- Likewise, the authors say that “Data collection took place for one minute using the Moya Ramon protocol.” Is the protocol proper to compare HRV alteration in ALS patients?
Thank you for your comment. Indeed, the study aims to determine the suitability and effectiveness of this type of technology, much faster and non-invasive, applied in ALS patients to identify the previously reported HRV alterations in this disease, which is the main goal of the study. To make this aspect clearer, the aim of the article has been further elaborated.
3.Is one minute interval enough to perform frequency-domain HRV analysis without any special processing technique? Generally, in order to measure power for LF band (0.04 to 0.15) within a short interval less than five minutes, a highly skilled analysis, as known as (ultra) short term HRV analysis, is required. This maybe the reason the author failed to find significant difference in frequency domain analysis using LH/HF ration, which can clearly shows the balance of the Sympathetic Nervous System (SNS) and the Parasympathetic Nervous System (PNS).
The Moya Ramón protocol used in our study employs shorter sample collection times than usual, specifically 1 minute. The Luong A. 2021 protocol also refers to a measurement time of 1 minute for high frequencies (HF) and 2 minutes for low frequencies (LF). Therefore, sample collection times have been considerably reduced over time; hence, in our study, and given the complexity of managing ALS patients, we decided to apply this 1-minute collection protocol, comparing the results with those obtained by other authors who used measurements over longer periods. As it can be concluded from the discussion, our results were very similar to those obtained by other authors (Silveira et al. (28) who measured for 10 minutes, or by Iscan et al. (15) who applied a 24-hour Holter electrocardiogram). Moreover, our results regarding the LF/HF Ratio variable, as commented on by the reviewer, also coincide with those obtained in previous studies (references 32, 33, and 34), reaffirming that the time employed may be sufficient.
Reviewer 2 Report
Comments and Suggestions for Authors
This study used a Polar H7 Bluetooth device connected to a chest belt to acquire RR data for one minute.
Some comments are needed.
The authors reviewed Billman´s paper named: The LF/HF ratio does not accurately measure cardiac sympatho-vagal balance, reference [18]. Nevertheless, they state that this ratio allows for the non-invasive evaluation of the autonomic cardiac control balance, a higher ratio indicating sympathetic predominance.
Also, they claim that the differences found in the several indexes reported between ALS and healthy persons reflect that the ANS i s not working correctly.The authors could support the notion that the ANS expresses stressful adaptations that let the patients survive.
The observation that heart rate declines, despite increases in sympathetic nerve activity, highlights the complex non-linear interactions of the sympathetic and parasympathetic nervous system. [18]
They have very high dispersion results, suggesting that their population group is very heterogeneous and that it is not possible to characterize it with one-minute recordings. A personalized approach could be better for so many diverse disease conditions.
A personalized approach could be better for so many diverse disease conditions.
Author Response
Dear editor and reviewers, the authors sincerely appreciate the thorough and detailed review of the text. Below, we present the improvements made following your detailed instructions
Reviewer 2
The authors presented the work titled as "Analysis of Heart Rate Variability in Individuals Affected by Amyotrophic Lateral Sclerosis."
There are some comments which need to be addressed:
This study used a Polar H7 Bluetooth device connected to a chest belt to acquire RR data for one minute.
Some comments are needed.
The authors reviewed Billman´s paper named: The LF/HF ratio does not accurately measure cardiac sympatho-vagal balance, reference [18]. Nevertheless, they state that this ratio allows for the non-invasive evaluation of the autonomic cardiac control balance, a higher ratio indicating sympathetic predominance.
We find your comment very interesting. It is true, as the reviewer points out, that in Billman's opinion article, there is doubt about whether LF/HF is an accurate measurement of autonomic cardiac control balance. However, according to the review by Junichiro Hayano and Emi Yuda in 2021, which we included in the article [23], it does seem to be suitable for determinations over a short period of time, as it is the case in our study.
Also, they claim that the differences found in the several indexes reported between ALS and healthy persons reflect that the ANS is not working correctly. The authors could support the notion that the ANS expresses stressful adaptations that let the patients survive.
Thank you very much for your contribution. In response to your suggestion, we have added that to the discussion, and we believe it substantially improves the work.
The observation that heart rate declines, despite increases in sympathetic nerve activity, highlights the complex non-linear interactions of the sympathetic and parasympathetic nervous system. [18]
They have very high dispersion results, suggesting that their population group is very heterogeneous and that it is not possible to characterize it with one-minute recordings. A personalized approach could be better for so many diverse disease conditions.
Indeed, we agree with the reviewer that there is considerable dispersion in the results obtained. In fact, we decided to integrate all subjects into a single group in order to conduct more robust analyses and establish benchmarks. This is a limitation that we have added to the study's limitations. Once again, we appreciate your contribution.